# Differential Immunomodulatory Effects of Head and Neck Cancer-Derived Exosomes on B Cells in the Presence of ATP

**DOI:** 10.3390/ijms232214446

**Published:** 2022-11-21

**Authors:** Mali Coray, Veronika Göldi, Lukas Schmid, Laura Benecke, Fabrício Figueiró, Laurent Muller

**Affiliations:** 1Department of Biomedicine, University of Basel, 4031 Basel, Switzerland; 2Department of Otolaryngology and Head & Neck Surgery, University Hospital Basel, 4031 Basel, Switzerland; 3Departamento de Bioquímica, Instituto de Ciências Básicas da Saúde, Universidade Federal do Rio Grande do Sul, Porto Alegre 90035-003, Rio Grande do Sul, Brazil

**Keywords:** tumor-derived exosomes, HNSCC, exosomes, extracellular vesicles, B cells, regulatory B cells, ATP, adenosine, immunomodulation, apoptosis

## Abstract

Head and neck squamous cell carcinoma (HNSCC) is an aggressive malignancy. Tumor-derived exosomes (TEX) have immunoregulatory properties. Adenosine triphosphate (ATP) and its immunosuppressive precursor adenosine (ADO) have been found in cancerous tissue. We investigated the effect of TEX on B cells in the presence of ATP. TEX were isolated from human HNSCC cell line (PCI-13) cultures and co-cultured with peripheral blood B cells of healthy donors, with or without TEX in different concentrations and with or without a low (20 µM) or high (2000 µM) ATP dose. We were able to demonstrate that TEX inhibit B-cell proliferation. The addition of TEX to either ATP concentration showed a decreasing trend in CD39 expression on B cells in a dose-dependent manner. High ATP levels (2000 µM) increased apoptosis and necrosis, and analysis of apoptosis-associated proteins revealed dose-dependent effects of ATP, which were modified by TEX. Altogether, TEX exhibited dual immunomodulatory effects on B cells. TEX were immunosuppressive by inhibiting B-cell proliferation; they were immunostimulatory by downregulating CD39 expression. Furthermore, TEX were able to modulate the expression of pro- and anti-apoptotic proteins. In conclusion, our data indicate that TEX play an important, but complex, role in the tumor microenvironment.

## 1. Introduction

Head and neck squamous cell carcinoma (HNSCC) is the sixth most common cancer worldwide. It bears one of the strongest impacts on quality of life and the worst prognosis, with high mortality and morbidity [1,2,3,4]. Unlike other solid tumors, HNSCC is regarded as highly immunogenic and, thus, susceptible to immunotherapeutic approaches [5]. Amongst lately developed checkpoint inhibitors such as PD-1/PD-L1 inhibitors, new immunotherapeutic approaches in anti-cancer therapy are desperately needed to prevent unfavorable outcomes [2].

As tumor evasion from the host immune system is a well-recognized phenomenon, a more in-depth understanding of the tumor-promoting microenvironment and its immunosuppressive properties is of utmost importance [6,7,8]. Amongst the innumerable molecules and immune cells that are present in the pro-inflammatory and hypoxic environment of human solid tumors, we aim to highlight the interactions of B cells and tumor-derived exosomes (TEX) in an environment where unusually high levels of extracellular adenosine triphosphate (ATP) and adenosine (ADO) are measured [9].

ATP is found ubiquitously throughout the body in all cells. Intracellular ATP is best known as the body’s energy currency. Extracellular ATP (eATP) is widely considered a damage-associated molecular pattern (DAMP) or a signal transducer. Increasing eATP levels signal cellular stress, injury, or death to activate inflammatory and reparatory responses. Pathologically high levels of eATP are found in the presence of malignancy, inflammation, and ischemia [9,10,11,12,13,14,15]. In the tumor microenvironment (TME), ATP induces inflammation and influences the initial tumor establishment, progression, dissemination, and anti-tumor immunity [14,15,16]. However, ATP is reported to play a dual immunomodulatory role in the TME, depending on the ATP concentration, the function of the recruited immune cell, the expression pattern of the ATP receptor subtype, and the presence of ATP-degrading ectoenzymes [15].

In contrast, ADO—a product of ATP hydrolysis—prevents excessive inflammation, impairs the immune system, and favors tumor progression [12,17]. High levels of ADO are also found in the TME and other inflammatory sites [15]. Four different G protein-coupled receptors are described [18]. Tumor cells also express ADO receptors. In HNSCC cells, the ADO receptor A2B (ADORA2B) has been detected [19,20]. Upon inhibition of ADORA2B, Wilkat et al. observed a significant decrease in the proliferation of human HNSCC cells, which supports the new therapeutic approach to modulate the ADO pathway [21]. Phase I clinical studies for a selective ADORA2B antagonist in combination with a PD-1 antibody have been tested on solid tumors [22].

The production of ADO is accomplished in two steps by extracellular enzymes linked to the cell membrane: CD39 (ectonucleotide triphosphate diphosphohydrolase) promotes ATP and ADP hydrolysis; CD73 (ecto-5′ nucleotidase) promotes AMP to ADO hydrolysis [11,12,23,24]. Noteworthily, eATP is the main source of ADO generation in the TME [15]. Enzymatic production can occur by tissue cells, including tumor cells, as well as immune cells (e.g., regulatory B cells, inducible regulatory T cells, myeloid-derived suppressor cells) that express CD39 and CD73 on their cell surface [16,23,25,26,27]. It is well-established that these enzymes promote tumor progression in different cancer types (e.g., HSNCC, breast cancer, intracranial neoplasia) [17,28,29,30].

Moreover, exosomes are found in the TME. Interactions between tumor cells and the immune system via exosomes have been observed [31,32]. These membrane-bound vesicles (30-150nm in diameter) are present in all body fluids. Originally, they were considered merely artifacts or fragments of dead cells. However, it has been discovered that they carry a broad array of bioactive molecules (nucleic acids, proteins, lipids) [33,34]. Accumulating evidence supports the pivotal role of tumor-derived exosomes (TEX) in tumor progression, angiogenesis, and immune escape [35].

In cancer, extraordinarily high amounts of exosomes are produced. It was described that in the TME, TEX led to the induction of apoptosis in effector cells, such as cytotoxic T cells, natural killer cells, and dendritic cells through upregulation of FasL, granzyme B, perforin IL-10, TGF-β1, and CTLA-4. Intriguingly, exosomes also carry and release CD39 and CD73, which play a major role in immunosuppression [25,27,31,32,36,37].

At the same time, the presence of major histocompatibility classes I and II, co-stimulatory molecules, and tumor-associated antigens on the surface suggests anti-tumor and immunostimulatory activities [35,38,39]. Immunostimulatory properties of TEX, such as the induction of cytotoxic T cells in antigen-based cancer immunotherapy, have been described. Further, changes in the expression levels in immunostimulatory genes in T cells have been observed upon co-incubation with TEX [38,40,41].

Altogether, a dual role of TEX, with immunoinhibitory as well as immunostimulatory characteristics, has been increasingly considered in the complex TME [38].

A significant contribution of regulatory immune cells to tumor progression has been widely reported. Whereas the effects of TEX on regulatory T cells (Tregs) have been researched extensively [32,42,43], their effects on B cells and their regulatory subsets have yet to be elucidated.

Regulatory B cells (Bregs) are a newly described subset of B cells that have been shown to play a pivotal role in regulating immune responses through the suppression of various cell subsets, including T effector cells. Through secretion of anti-inflammatory cytokines such as IL-10, this newly described B-cell subset shows immunosuppressive properties. They are characterized by high expression levels of CD39 on their cell surface (CD39+ high B cells), known to attenuate anti-tumor immune responses [25,44,45,46,47]. A recently published study by Jeske et al. described the presence of these ADO-producing Bregs in human and murine HNSCC. By suppressing the B-cell receptor signaling pathway, they demonstrated the inhibitory effect of ADO on effector B cells. Further, reduced proliferation of activated B cells in the presence of ADO was observed [22]. Like Bregs, exosomes process ATP to the immunosuppressive ADO by the activity of the ectonucleotidases CD39 and CD73 expressed on their surface. An inhibitory effect of exosomes on B-cell receptor signaling has been demonstrated by Schroeder’s group. However, the inhibitory effect of exosomes on the function of healthy B cells was comparable between exosomes from healthy individuals and HNSCC patients, suggesting a physiological B-cell inhibitory role [48].

Altogether, improved knowledge of the TME and its interaction with the immune system through various molecular pathways is valuable in assessing novel opportunities for immunomodulatory cancer therapeutics. In this study, we aim to evaluate the effect of HNSCC-derived exosomes in the presence of high ATP levels on B cells.

## 2. Results

### 2.1. Isolation of Tumor-Derived Exosomes (TEX) and Regulatory B Cells (Bregs) from HNSCC

As a proof of concept, TEX and Bregs were isolated from HNSCC. To separate the effects of exosomes deriving from cancerous cells from those of exosomes deriving from non-cancerous cells, exosomes from a single HNSCC cancer source (PCI-13 cell line) were used for the experiments. TEX were isolated and enriched through a combination of filtration, size-exclusion chromatography, and differential centrifugation. They were quantified using a BCA protein assay and characterized by transmission electron microscopy (TEM) (Figure 1a). ZetaView^®^ was used to characterize these particles by their size and concentration (Figure 1b). The presence of the general exosome marker CD81 further verified the presence of exosomes (Figure 1c). As such, we ensured that we isolated exosomes and no other extracellular vesicles or contaminants.

To evaluate the presence of Bregs in HNSCC, biopsies from HNSCC patients were dissociated and stained for CD20, CD39, CD24, and CD38. Albeit in small percentages, Figure 1d shows that Bregs could be isolated from the TME.

The results are presented as the mean of three independent experiments with their standard deviation. As previously described by us and other colleagues, we looked separately for the CD39high and the double-positive CD24+CD38+ populations after gating for a general B cell marker (e.g., CD20+ cells) [25,49,50]. In the following experiments, B cells were isolated from healthy donors as described in the Methods section. Flow cytometry was used to assess the purity of isolated B cells (>90%).

### 2.2. Effects of TEX and ATP on B-Cell Proliferation

#### 2.2.1. Immunosuppressive Effects of TEX on B-Cell Proliferation

Activation of B cells led to increased cell proliferation (Figure 2). Isolated B cells from healthy donors were either kept in a resting state (Brest) or activated with CD40L and IL-4 (Bact). The time of analysis was chosen according to the flow-cytometry-based CFSE histogram data. Determining factors were multiple peaks as a sign of proliferation in the positive control (Bact) and a singular peak representing the parental generation in the negative control (Brest) (Figure 2a). Figure 2a also illustrates that adding TEX (10 µg/mL) to Bact decreased B-cell proliferation in the CFSE assay.

#### 2.2.2. Dose-Dependent Immunomodulatory Effects of ATP and TEX on B Cells

ATP can be converted by B cells or TEX to ADO. ADO is a strong immunosuppressive agent that plays a pivotal role in the TME [47,51,52]. To test the impact of ATP on B cells in a physiological and pathological state, two extreme ATP concentrations were chosen as an educated guess for our model. Hereby, 20 µM of ATP represented the eATP presence in physiological conditions. The amount of 2000 µM of ATP represented the eATP concentration in the TME. Hence, resting or activated B cells were co-cultured with 20 µM or 2000 µM ATP for 96 h. Figure 2b shows the relative proliferation index compared to Brest (value = 1) in the flow-cytometry-based CFSE assay. Lower values mean less CFSE fluorescence compared to Brest and, as such, proliferating cells. The mean of four independent experiments is shown on the heatmap.

In Brest, changes in proliferation did not reach statistically significant values (Brest; upper part of Figure 2b). In Brest, the highest increase in proliferation occurred when co-incubated with TEX (10 µg/mL) and 20 µM ATP.

Bact depicted in the lower part of Figure 2b proliferated as expected (Brest vs. Bact: *p* ** = 0.0046; *t*-test, two-tailed). Bact with 20 µM ATP showed no significant change in proliferation levels. Additionally, further addition of TEX did not elicit significant results. The most striking difference was observed in Bact co-incubated with 2000 µM ATP (Bact vs. Bact + 2000 µM ATP: *p* *** = 0.0003, Dunnett’s multiple comparisons test): 2000 µM ATP clearly reduced B-cell proliferation. Adding TEX to Bact + 2000 µM ATP was significantly different from Bact (Bact vs. Bact 2000 µM ATP + TEX: *p* * = 0.0334, Dunnett’s multiple comparisons test), but not from Bact + 2000 µM ATP without TEX.

### 2.3. Effects of ATP and TEX on CD39 Expression on B Cells

#### 2.3.1. Dose-Dependent Effects of ATP on CD39 Expression on B Cells

High CD39 ectoenzyme expression in B cells translates into high immunosuppression as previously shown [10]. Figure 1d demonstrates the presence of Breg subsets in the TME. As a next step, we investigated the influence of exogenously added ATP on CD39 expression on B cells. CD39 surface expression was evaluated by flow cytometry. Hereby, six independent experiments were performed.

Figure 3a displays the effect of low and high ATP concentrations on CD39 expression on B cells after activation with CD40L and IL-4. The mean of the two ATP interventions (20 µM vs. 2000 µM) relative to Bact is plotted in a box plot graph. Here, 20 µM ATP did not alter the overall CD39 expression in Bact. However, 2000 µM ATP significantly decreased the expression of CD39 in Bact (Bact vs. Bact + 2000 µM ATP: *p* * = 0.014; Bact + 20 µM vs. Bact + 2000 µM ATP: *p* * = 0.012).

In Figure 3b, the representative flow cytometry analysis illustrates a clear increase in CD39 surface protein expression following B cell activation (left), but a decrease in CD39 expression on Bact co-incubated with 2000 µM ATP (right).

Regarding the CD39high B-cell subset (Figure 3c), a highly significant increase to about 5% was observed upon the activation of B cells (Brest vs. Bact: *p* **** < 0.0001). This was comparable to our previous experiments [25]. In the presence of 2000 µM ATP, but not 20 µM, these immunosuppressive B cells significantly decreased (Figure 3c, Bact vs. Bact + 2000 µM ATP: *p* * = 0.047).

#### 2.3.2. Dose-Dependent Effect of TEX on CD39 Expression on B Cells in Presence of ATP

The dose-dependent effect of TEX on the overall CD39 expression on B cells in the presence and in the absence of ATP is presented in Figure 4a. Activation of B cells significantly increased CD39 expression on B cells (Brest vs. Bact: *p*****< 0.0001). Adding 20 µM ATP did not alter CD39 expression. However, 2000 µM ATP significantly reduced CD39 expression (Bact vs. Bact + 2000 µM ATP: *p*** = 0.0026). Within the individual groups, the influence of TEX was observed as a reduction in CD39 expression in a dose-dependent manner. Only in Brest, TEX did not impact CD39 expression. Adding the highest TEX concentration (20 µg/mL) to Bact in the presence of 2000 µM ATP lowered the level of CD39 expression close to the range detected on Brest. In Figure 4b, the dose-dependent impact of TEX on the CD39high B-cell subset in Brest, Bact, Bact + 20 µM ATP, and Bact + 2000 µM ATP is presented. With the exception of Brest, a dose-dependent reduction in the CD39high B-cell rate through TEX was observed. This effect was enhanced by ATP. At 20 µg/mL TEX, the CD39high rate in Bact with or without ATP dropped to almost the range measured in Brest. By comparing the Bact settings (0, 20 or 2000 µM ATP) to Brest, the difference between each Bact setting and Brest lost its significant value at 20 µg/mL TEX. (With 0 µg/mL TEX: Brest vs. Bact: *p* *** = 0.0003; Brest vs. Bact + 20 µM ATP: *p* **** < 0.0001; Brest vs. Bact + 20 µM ATP *p* **** < 0.0001. With 5 µg/mL TEX: Brest vs. Bact: *p* *** = 0.0023; Brest vs. Bact + 20 µM ATP: *p* * = 0.0237; Brest vs. Bact + 20 µM ATP: *p* * = 0.0323. With 10 µg/mL TEX: Brest vs. Bact: *p* *** = 0.0067.)

### 2.4. Effects of ATP and TEX on Cell Death in B cells

#### 2.4.1. Effects of ATP on Annexin V/Propidium Iodide (PI) Positivity in Activated B Cells

In our previous flow cytometry experiments, an increase in debris in the forward and side scatter images was noticeable, especially in the 2000 µM ATP condition. Hence, the effects of ATP on the apoptosis and necrosis of B cells were analyzed. Hereby, isolated B cells from healthy donors were activated and treated with a low (20 µM) or high (2000 µM) ATP dose. The cells were stained with annexin V-FITC/propidium iodide (PI) and measured by flow cytometry. The acquired flow cytometry data are presented in Figure 5a. The means of three independent experiments are shown in Figure 5b. Figure 5a (upper panel) and Figure 5b show that co-incubation of Bact with ATP—particularly the high dose of 2000 µM—reduced the B cell viability and increased the rate of late apoptotic cells. (viable: Bact vs. Bact + 20 µM ATP: *p* * = 0.0384; Bact vs. Bact + 2000 µM ATP: *p* ***** < 0.0001; Bact + 20 µM ATP vs. Bact + 2000 µM ATP: *p* **** < 0.0001; late apoptotic: Bact vs. Bact + 2000 µM ATP: *p* ***** < 0.0001; Bact + 20 µM ATP vs. Bact + 2000 µM ATP: *p* ****< 0.0001).

#### 2.4.2. Effects of ATP and TEX on Annexin V/Propidium Iodide (PI) Positivity in Activated B Cells

Since TEX can rapidly metabolize ATP, we analyzed their influence on annexin V and PI expression in B cells with low (20 µM) and high (2000 µM) ATP concentrations. Figure 5a (lower panel) and Appendix A show that adding TEX (10 µg/mL) did not alter the annexin V/PI fluorescence in Bact alone or in either ATP concentration.

### 2.5. Effects of ATP and TEX on Apoptosis-Associated Proteins in B Cells

#### 2.5.1. Overall Effects of ATP and TEX on Apoptosis-Associated Proteins in Activated B Cells

To improve the understanding of the undergoing mechanisms, dot blot experiments to study 43 different apoptosis-associated proteins were performed. Hereby, the effects of two ATP conditions (20 µM and 2000 µM) with or without TEX (10 µg/mL) on the expression of these apoptosis-associated proteins in Bact after 96 h co-incubation were analyzed. Four independent experiments of isolated B cells were performed. Protein expression was evaluated by the chemiluminescent signal on the dot blots (Appendix A).

In Figure 6a, the means of the apoptosis-associated proteins are presented in a heatmap with unsupervised clustering. The six B-cell interventions are shown on the x-axis, and the 43 proteins are listed on the right y-axis. The graph was generated using the web-based heatmapper (http://www.heatmapper.ca/expression/; accessed on 27 January 2021) developed by Babicki et al. [53]. Unsupervised clustering of the columns separated the six conditions into two groups: the green-dominated group of Bact, including the low ATP concentration (with or without TEX), and the pink-dominated group of Brest, including Bact with the high ATP concentration (with or without TEX). In other words, modulation of the apoptotic proteins in Bact + 2000 µM ATP more closely resembled Brest than Bact. Bact + 2000 µM ATP, with the strongest intensity of pink color, differed from Bact + 20 µM ATP, which displayed the most intense green color. The two TEX interventions clustered on both ends of the heatmap. They separated as a distinct cluster from the samples without TEX, especially Bact + 20 µM ATP + TEX. The area between SMAC and IGFBP-3 in the protein profile of Bact + 2000 µM ATP + TEX stood out with its white to green colors amidst the pink group. For these proteins, the modulation more closely resembled that of the green group than that of the pink group, especially Bact.

Overall, Survivin was one of the proteins that exhibited the greatest difference between the two groups. In the green group, increased Survivin expression was observed. In the pink group, decreased Survivin expression was observed.

Figure 6b shows a heatmap of the expression of the 43 apoptosis-associated proteins as an overall mean on a Pearson correlation matrix with their respective r-values and statistical significances (*p*-values). Despite some variations between each experiment, highly significant differences between each group were observed. The most striking differences appeared between Brest and Bact or Bact + 20 µM ATP +/− TEX, which is also seen in Figure 6a.

#### 2.5.2. Effects of ATP and TEX on Pro- and Anti-Apoptotic Proteins in Activated B Cells

Appendix A shows the same values as Figure 6a. However, the results are divided into pro- and anti-apoptotic proteins in a semi-supervised heatmap. Again, the low ATP dose clustered into the Bact group. The high ATP dose clustered into the Brest group. Notably, TEX addition to Bact + 20 µM ATP or Bact + 2000 µM ATP further modulated the expression of the proteins. Overall, the strongest expression was seen in Survivin and p21, as well as Caspase 3, Fas, and IGFBP-5.

Appendix A displays the heatmaps of the expression of the anti- and pro-apoptotic proteins as an overall mean on a Pearson correlation matrix with their respective r-values and statistical significances (*p*-values). Differences between each group were highly significant. For the anti-apoptotic proteins, the biggest differences were seen between Brest and Bact or Bact + 20 µM ATP +/− TEX. For the pro-apoptotic proteins, these differences were less pronounced. Instead, the differences were greater between Bact + 20 µM ATP + TEX and Brest or Bact + 2000 µM ATP +/− TEX. The matrix of the anti-apoptotic proteins showed lower r-values than that of the pro-apoptotic proteins (see legend bar), which translates into a greater difference between the groups.

#### 2.5.3. Effects of ATP and TEX on Specific Apoptosis-Associated Proteins in Activated B Cells

Based on the prominent results in Figure 6a and the key functions in the common apoptotic signaling pathway, apoptosis-associated proteins were selectively analyzed.

This in-depth analysis revealed modulatory effects of B-cell activation, ATP, and TEX on proteins of the intrinsic, extrinsic, and p53-mediated apoptotic pathways. 

The markers for the intrinsic (mitochondrial) pathway are the pro-apoptotic proteins BIM, Bax, SMAC, and HTRA, and the anti-apoptotic proteins Bcl-2, Bcl-2, XIAP, Survivin, and Livin. The markers for the extrinsic (death receptor) pathway comprise FasL, Fas, and Caspase 8. Both pathways lead to activation of Caspase 3, which ultimately induces cellular apoptosis. Crosstalk between the extrinsic and intrinsic pathways is enabled by BID activation via Caspase 8 [54]. The tumor suppressor protein p53 is also known to induce pro-apoptotic Fas and Bax, as well as anti-apoptotic p21 [55]. Anti-apoptotic Hsp60 and Hsp70 block apoptosis via inhibition of p53, and various proteins of the intrinsic and extrinsic pathways [56,57,58]. Figure 7a demonstrates that B-cell activation led to nonsignificant upregulation of most of the selected pro- and anti-apoptotic proteins. Significant upregulation was observed especially for Survivin, Caspase 3, and p21. Adding 20 µM ATP to Bact only had a small impact on the expression of the selected apoptosis-associated proteins in Bact (Figure 7b). Further addition of TEX did not alter the expression of the apoptotic proteins (Appendix A). However, the addition of 2000 µM ATP to Bact led to a downregulation of all selected anti- and pro-apoptotic proteins (Figure 7b). Significant downregulation was observed for Bax, HTRA, Survivin, Livin, Fas, Caspase 3, p21, and Hsp60. Further addition of TEX counteracted the effect of 2000 µM ATP and caused a nonsignificant upregulation of almost all apoptotic proteins (Figure 7c). Significant changes were seen in Survivin and Livin. For all interventions, Survivin consistently displayed the highest significant upregulation of all apoptotic proteins (Figure 7a–c). The findings for Bax were confirmed by Western blotting (Appendix A).

## 3. Discussion

Accumulating evidence supports tumor-derived exosomes’ (TEX) pivotal role in tumor progression and immune escape [31,35]. These vesicles of 30–150 nm size carry a broad array of bioactive molecules on their bilayer surface and represent a way of communication between the tumor and its environment [34,59].

In the first figure, we checked our isolates using TEM and nanoparticle tracking analysis. The morphology and size distribution were within the range of exosomes, and we were able to detect the presence of the common exosome marker CD81 in our isolates.

It has already been observed that TEX have a significant effect on T cells [32,42,43]. However, the impact on B cells is less well-known and deserves further attention. B cells are central effector cells of the adaptive immune response [32,42,46]. Their implication in cancer as important players has progressively been recognized. Infiltration of B cells in the TME has been detected and is associated with better survival [60].

In our study, we wanted to highlight the role of the recently described CD39high B cell subset. These cells are thought to be of utmost importance in regulating immune responses through the suppression of various cell subsets, including T effector cells. This newly described B-cell subset shows immunosuppressive characteristics by secretion of anti-inflammatory cytokines such as IL-10, inhibition of T effector cell proliferation, and activation. Further, this Breg subset is characterized by high expression levels of the enzyme CD39 on its cell surface (CD39high B cells). Its exact role in tumor promotion, however, is not fully understood yet [25,44,46].

In our biopsy samples from HNSCC patients, we were able to isolate Breg subsets. Not only the better-known CD24high/CD38high B cells, as described by Lechner et al. [46], but also the less-known CD39high B-cell subset could be measured in our tumor dissociates. This indicates that the CD39high Breg subset is part of the tumor-infiltrating lymphocytes and could play an essential role in the cancer niche.

Upon in vitro co-incubation of Bact with TEX, we were able to observe an apparent inhibition of proliferation, suggesting immune inhibitory properties of TEX (Figure 2). Similar findings were reported by Schroeder et al. [48]. However, there are some differences to our study. In our experimental setup, TEX and the activators were added simultaneously to the B-cell culture. The isolated exosomes by Schroder et al. were plasma-derived, i.e., the cellular source of the exosomes is unknown. Subsequently, observed effects cannot be correlated to exosomes of a specific cell origin (TEX vs. immune/other cell-derived exosomes). For instance, it was concluded that exosomes from healthy controls did not differ significantly from those of HNSCC patients. This finding could, however, be explained by the non-tumor-derived exosomes present in the plasma or common exosomal markers. Moreover, the exosomal profile likely varies extensively from patient to patient. For these reasons, we used cell-culture-derived exosomes that were derived from a single head and neck cancer cell line (PCI-13).

eATP acts as a signal transducer for cellular damage or stress. In the TME, the high eATP levels influence the initial tumor establishment, progression, dissemination, and anti-tumor immunity [14,15,16]. To mimic the pathologically high levels of eATP found in the TME and the lower, more physiological ATP levels, we used high (2000 µM) and low (20 µM) ATP concentrations for our experimental setup.

Thus, the co-cultures were repeated in the presence of eATP. The results are shown in Figure 2b: 2000 µM ATP inhibited proliferation. However, 20 µM ATP did not significantly impact proliferation in Bact. The inhibitory effect of TEX on B-cell proliferation seen in the absence of ATP (Figure 2a) was not apparent in the presence of either ATP concentration.

Cancer cells, but also immune cells, carry purinergic type 2 receptors (P2R) that are able to sense eATP [15,16,61]. Depending on the ATP concentration and activated purinergic receptor subtype, proliferation, proliferation inhibition, or cell death is induced [16,29,62]. Many of the P2R subtypes are activated by ATP in the low micromolar ranges. The P2X7 receptor, however, requires a high ATP concentration, as it has a high activation threshold (≥500 µM) [15,51,63,64]. P2X7 activation causes a non-selective pore formation for molecules of up to 0.9 kDa, the release of pro-inflammatory cytokines, the production of free radicals, a large-scale ATP release to further activate purinergic signaling and inflammation, and the induction of cell death [51,61]. In cancer, P2X7 activation has displayed both immunostimulatory and -suppressive properties depending on the cell type and the level of activation [29,61]. In B cells, Pupovac et al. described that 1000 µM ATP activated P2X7, which led to the shedding of CD23, a low-affinity IgE receptor [52]. Our results of proliferation and cell death suggest that the high level of 2000 µM ATP led to P2X7 activation, which favored cell death over proliferation. Still, this remains speculative and cannot be answered in our experiments.

In the TME, the net effect of eATP depends not only on its concentration or the expression pattern of purinergic receptor subtypes, but also on the presence of ATP-degrading ectoenzymes. ATP can be rapidly metabolized by the membrane-bound ectonucleotidases CD39 and CD73 to highly immunosuppressive ADO. It has been reported that ADO strongly inhibited B-cell proliferation [29]. Although high eATP is primarily associated with a heightened anti-tumor response, eATP can indirectly elicit immunosuppressive effects via ADO accumulation. In the TME, eATP is the main source of ADO. Thus, continuous eATP supply in the TME could actively be generated in the interest of the tumor. In line with that, the accumulation of ADO could alternatively explain the observed anti-proliferative impact of 2000 µM ATP on B cells seen in Figure 2b [15,16,29]. The ATP-degrading ectoenzyme CD39 is expressed on cancer cells and endothelial cells, but also various immune cells, including B cells [15,22,23,25]. It was indeed found that B cells displayed a high ATP-degrading activity [29]. Hence, we tested the CD39 expression on B cells and its modulation by ATP and TEX.

Activation of B cells significantly increased the overall CD39 expression and the rate of the CD39high Breg subset (Figure 3b,c). Adding 20 µM ATP did not modulate these results. In contrast, 2000 µM ATP downregulated the overall CD39 expression and the CD39high B-cell rate (Figure 3). Hence, the high 2000 µM ATP concentration exerted immunomodulatory signals by reducing immunosuppressive B cells, whereas 20 µM ATP alone did not elicit any further changes to Bact.

Interestingly, TEX reduced the CD39 expression on Bact and the CD39high B-cell rate in Bact in a dose-dependent manner, especially in the presence of ATP (Figure 4). Thus, in the TME, the high TEX concentration modulated the ATP impact with regard to the CD39+ and CD39high B-cell populations.

Schuler et al. showed that ADO led to inhibition of the intracellular Bruton tyrosine kinase in effector B cells, but not in Bregs. This inhibition led to a downregulation of CD39 in effector B cells in vitro. Inhibition of the ADO 2A (A2A) receptor led to a significant decrease in tumor volume and an increase in B-cell infiltration [22]. With our current data, the role of the A2A receptor cannot be answered.

However, Whiteside et al. described that stimulation of the A2A receptor reduced exosome production by cancer cells, including an HNSCC cell line [65]. Thus, new targeted therapies that inhibit the A2A receptor could undesirably cause a significant increase in immunosuppressive exosomes. However, data on this topic are missing.

In immune cells, P2X7 has been shown to mediate the ATP-induced cell death pathways, including apoptosis and necrosis [16,62,63,66]. Hence, we investigated cell death—especially apoptosis—in our B cell/TEX/ATP model. Repeatedly, we noticed that 2000 µM ATP strongly increased cell death, measured by an increase in annexin V and PI signals, compared to 20 µM ATP (Figure 5). Adding TEX to either ATP concentration did not significantly change the annexin V and PI signals (Figure 5a (lower panel) and Appendix A). In conclusion, the ATP concentration was the decisive force in inducing apoptosis and necrosis in B cells in our model. Particularly with 2000 µM ATP, it is likely that the results were further enhanced by the vicious cycle of the continuous release of intracellular ATP into the extracellular space through the cell death induced by the high ATP levels.

eATP-P2X7 signals have been linked to the intracellular mammalian target of rapamycin (mTOR), the crucial regulator for basic cellular functions such as survival, proliferation, and apoptosis. eATP (≥1000 µM) was found to inhibit mTOR via upstream blockade of phosphatidylinositol-3-OH kinase/protein kinase B (PI3K/AKT) and activation of AMP kinase (AMPK) in cancer cells [63]. The classical energy sensor AMPK is known to be activated following an increase in the intracellular AMP/ATP or ADP/ATP ratio, which indicates low energy storage in the cell. The ratios were also found to be lower in dying cells compared to proliferative cells [63,67]. Indeed, it was observed that intracellular AMP and ADP increased, and intracellular ATP decreased following exposure to high eATP levels. This perturbation of intracellular adenine nucleotides was explained by the non-selective pore formation due to P2X7 activation. Ultimately, it was observed that the mTOR inhibition induced by the high eATP concentrations led to cell death [63].

To improve our understanding of the underlying mechanisms, we analyzed a panel of 43 apoptosis-associated proteins, including some proteins involved in the mTOR signaling pathway. The Pearson correlation matrix of the overall changes in the six conditions showed the biggest differences between Brest and Bact with/without 20 µM ATP. The lowest r-value was seen correlating Brest with Bact + 20 µM ATP + TEX, suggesting that TEX modulated the low ATP dose the most (Figure 6b). Looking at the clustered heatmap of the mean values of the protein expression, we observed two different groups, where the low and high ATP concentrations diverged considerably (Figure 6a). Additionally, TEX significantly modulated protein expression and presented a distinct pattern depending on the ATP concentration. These results show that TEX are indeed able to influence apoptosis-associated protein expression. For instance, TEX significantly increased the expression of Livin in Bact + 2000 µM ATP (Figure 7c). As Livin is an anti-apoptotic protein [68], TEX may counteract the pro-apoptotic signals delivered by high ATP doses to a certain extent.

There is some information on ATP-mediated cell death occurring via mTOR inhibition and the mitochondrial pathway [16,62,63,66]. Hereby, the inhibition of Bax was reported to reverse cell death [66]. We noted a significant decrease in Bax and Caspase 3 expression in Bact + 2000 µM ATP vs. the other interventions. The increase in cell death seen in Figure 5 suggests that also caspase-independent cell death occurred with 2000 µM ATP, such as colloido-osmotic lysis. This has also been described in macrophages with exposure to 3000 µM ATP [69]. The increase in PI signals observed with 2000 µM ATP (Figure 5) corroborates the loss of cell integrity, allowing the dye to enter the cell.

Another interesting finding is that Survivin showed a massively reduced expression in Bact + 2000 µM ATP. Survivin is usually upregulated in proliferating cells and cancer cells. It has also been described in tumor-derived exosomes [70], which were able to deliver Survivin to neighboring cancer cells and thus induce proliferation and deliver anti-apoptotic signals. In our results, TEX did not significantly influence these results.

However, differences in ATP sensitivity or response among B-cell subsets are anticipated by the varying expression patterns of P2R and ATP-metabolizing ectoenzymes. Among T cells, more cell death occurred in CD4+CD25+ Tregs than in CD4+/CD8+ effector T cells following high ATP levels. Hereby, the CD8+ T-cell subset was the most resistant [71]. For B cells, this has yet to be investigated.

To unravel the exact signaling pathways of ATP/ADO/TEX and the susceptibility of the B-cell subset to cell death, further investigations with isolated/sorted B-cell subsets, including different inhibitors of the ATP/ADO pathway, should be performed.

## 4. Materials and Methods

### 4.1. Cell Culture

The human HNSCC cell line (PCI-13) was kindly donated by Prof. Dr. T. Whiteside from the Hillman Cancer Institute, University of Pittsburgh, Pennsylvania, USA. The cells were cultured in DMEM (Gibco, Life Technologies, Carlsbad, CA, USA), supplemented with 10% fetal bovine serum (FBS, Gibco, Life Technologies, Carlsbad, CA, USA) and 1% penicillin/streptomycin (10,000 U/mL, Gibco, Life Technologies, Carlsbad, CA, USA). To diminish contamination of bovine extracellular vesicles, FBS was ultracentrifuged at 100,000× *g* for 2 h at 4 °C, discarded from its pellet, and then added to the culture medium.

### 4.2. Isolation of Exosomes

Exosomes were isolated from the supernatant of the cultured HNSCC cell lines according to an adapted previously described isolation protocol [64,72]. Following differential centrifugation (1000× *g* for 10 min at room temperature (RT), 3000× *g* for 10 min at 4 °C, and 10,000× *g* for 30 min at 4 °C), the supernatant was filtrated through a sterile 0.22 µm-pore filter (PES TOP, FilterBio Membrane Co., Ltd., Nantong City, Jiangsu, China) and concentrated using 100 kDa Amicon Ultra-15 mL Centrifugal Filter Devices (Sigma-Aldrich, St. Louis, MO, USA) by centrifugation. Next, size-exclusion chromatography was performed by placing 1 mL of the concentrated solution on a column (r = 1.5 cm) packed with Sepharose 2B gel (60–200 µm bead diameter, Sigma-Aldrich, St. Louis, MO, USA) to the height of 10 cm. The fourth and fifth exclusion volume fractions were kept for further analysis and experimentation, as they contained the purest yield of exosomes based on the characterization results.

### 4.3. Quantification and Characterization of Isolated Exosomes

BCA protein assays were performed according to the manufacturer’s instructions (#23225, Pierce BCA Protein Assay Kit, Thermo Fisher Scientific, Waltham, MA, USA) to determine protein concentrations for exosome quantification. To estimate the protein concentration, a curvilinear regression was established with bovine serum albumin. For the analysis of the morphology and size of the exosomes, transmission electron microscopy (TEM, Philips CM 100) was performed at the Biozentrum Basel. Briefly, the sample (5 µL) was placed on a copper disc. After 1 min, the copper disc was blotted with filter paper and washed thrice with droplets of dH2O (5 µL). The copper disc was stained twice with uranyl acetate 2% (5 µL) and then examined under the TEM. The size of the exosomes and their distribution were analyzed using the ZetaView^®^ nanoparticle tracking analyzer (Particle Metrix, Inning am Ammersee, Germany). To further verify the presence of the isolated exosomes, the immunoaffinity capture-based technique with anti-CD81 antibodies (#MA5-13548, Invitrogen, Thermo Fisher Scientific, Waltham, MA, USA) coupled via their Fc-region to magnetic Dynabeads^®^ Protein G (#10003D, Invitrogen, Thermo Fisher Scientific, Waltham, MA, USA) was employed. Briefly, magnetic Dynabeads^®^ with recombinant Protein G covalently coupled to their surface were incubated in a tube with anti-CD81 antibodies, with rotation for 1h at RT. After placing the tube on a magnet, the supernatant containing excess unbound antibodies was removed. After washing and resuspending the antibody-coated beads twice in PBS, the beads were incubated with the isolated exosomes for 3 h with rotation at RT. After washing twice with PBS, the exosome–antibody–bead complex was resuspended in PBS and imaged by TEM.

### 4.4. Peripheral Blood Mononuclear Cells (PBMCs) and Isolation of Peripheral Blood B-Cells

Isolation methods were previously described by us or other colleagues [22,25,47]. Briefly, buffy coats of healthy individuals obtained from the Swiss Red Cross Blood Donation Center Basel were centrifuged on Ficoll–Paque Plus (GE Healthcare Biosciences, Chicago, IL, USA) to separate PBMCs by a density gradient. To remove adherent cells, the collected PBMCs were incubated for 30 min at 37 °C in RPMI 1640 medium (#11875093, Gibco, Life Technologies, Carlsbad, CA, USA) (supplemented with 1% penicillin/streptomycin and 10% FBS depleted of extracellular vesicles) before the magnetic separation. Using anti-CD19-antibody-coated magnetic beads (#130-050-301, Miltenyi Biotec, Bergisch Gladbach, Germany), CD19+ B cells were isolated from the nonadherent PBMCs by positive selection according to the manufacturer’s instructions. The purity of the B-cell isolation was determined by flow cytometry with anti-CD20 (anti-CD20 FITC (#302304) or anti-CD20-PE (#302305), BioLegend, San Diego, CA, USA). Purity values were generally > 90%.

### 4.5. CFSE-Based Proliferation Assay and Co-Incubation Assays

Freshly isolated CD19+ B cells were stained with 5 µM CFSE (CFSE Cell Division Tracker Kit, #423801, BioLegend, San Diego, CA, USA) according to the manufacturer’s manual. CFSE is a stable, cell-permeable, non-fluorescent molecule. Intracellular esterases cleave the acetate groups of CFSE, producing fluorescent esters that are cell-impermeable. Proliferation is detected by a diminished CFSE fluorescence intensity. The ideal co-incubation duration for subsequent surface staining was determined by flow cytometry.

### 4.6. Co-Incubation Assays

CD19+ B cells were seeded at 200,000 cells/mL in RPMI 1640 medium (1% penicillin/streptomycin and 10% FBS depleted of extracellular vesicles) and co-cultured for 96 h as follows: only culture medium; TEX (5, 10, or 20 µg/mL); ATP (20 µM or 2000 µM; #R0441, Thermo Fisher Scientific, Waltham, MA, USA); TEX and 20 µM ATP, or TEX and 2000 µM ATP. During these 96 h, B cells were either kept in a resting state (resting B cells, Brest) or activated (activated B cells, Bact) with human IL-4 (200 IU/mL, #574004, BioLegend, San Diego, CA, USA) and CD40L (2 µg/mL, #591706, BioLegend, San Diego, CA, USA).

### 4.7. Cell Surface Antigen Staining for Flow Cytometry

Cultured CD19+ B cells were stained with a fixable viability dye (eFluor780, eBioscience, San Diego, CA, USA) in PBS for 30 min at 4 °C in the dark. For surface antigen staining, the cells were incubated with fluorochrome-conjugated antibodies (all from BioLegend, San Diego, CA, USA: anti-CD20-FITC, #302304; anti-CD20-PE, #302305; anti-CD39-BV421, #328214; anti-CD24-PE, #311106; anti-CD38-APC, #356606; or the respective isotype control) for 30 min at 4 °C in the dark. After washing and resuspending the cells in FACS Buffer (PBS + 2% FBS), the cells were analyzed using a CytoFLEX flow cytometer (Beckman Coulter, Brea, CA, USA). To compensate for spillover effects, bead-based single stainings were performed. Dead cells and debris were excluded from the analysis based on viability dye fluorescence and scatter signals. The CD39high B-cell subset was defined as previously described [25]. Appendix A shows the gating strategy. All the collected data were evaluated using the FlowJo Software version 10.6.2 for Mac.

### 4.8. Annexin V/Propidium Iodide (PI) Staining Assay

Apoptosis was determined by annexin V and PI staining (#640905 and #421301, BioLegend, San Diego, CA, USA) according to the manufacturer’s protocol. After washing cells in PBS and annexin V binding buffer (#422201, BioLegend, San Diego, CA, USA), the cells were incubated in an annexin V/PI staining solution for 10 min at RT in the dark. The cells were analyzed using a CytoFLEX flow cytometer (Beckman Coulter, Brea, CA, USA) and FlowJo Software version 10.6.2 for Mac.

### 4.9. Apoptotic Protein Profiling

Apoptotic protein profiling was conducted according to the manufacturer’s instructions for the Human Apoptosis Antibody Array (#ab134001, Abcam, Cambridge, UK). Hereby, the protein was extracted from the B cells after co-incubation, as mentioned above, using 2X Cell Lysis Buffer and supplemented with a Protease Inhibitor Cocktail. Protein levels were quantified by performing BCA protein assays (#23225, Pierce, Thermo Fisher Scientific, Waltham, MA, USA). The chemiluminescence signal was detected by the FUSION Pulse imaging system. The internal controls (positive and negative) showed the appropriate signal in each experiment. Representative dot blots are presented in Appendix A. Densitometry data from the array points were obtained using Image J. Prior to comparative analysis with GraphPad Prism version 9.0.0, data were normalized according to the manufacturer’s guidelines. A heatmap with unsupervised clustering was generated using the web-based heatmapper (http://www.heatmapper.ca/expression/; accessed on 27 January 2021) developed by Babicki et al. [53].

### 4.10. Western Blotting

Protein was extracted by lysing cells in RIPA Lysis and Extraction Buffer (#89900, Pierce, Thermo Fisher Scientific, Waltham, MA, USA). The protein concentrations were determined using BCA protein assays performed according to the manufacturer’s instructions (#23225, Pierce, Thermo Fisher Scientific, Waltham, MA, USA). The samples with equal protein contents and volumes were mixed with Laemmli sample buffer (end concentration of 1X, #1610747, Bio-Rad, Hercules, CA, USA) and boiled for 10 min at 95 °C at 1000 rpm. Subsequently, the proteins were separated by electrophoresis at 180 V using a 4–12% gradient gel (NuPAGE Bis-Tris, #NP0321BOX, Thermo Fisher Scientific, Waltham, MA, USA). Next, the gel was cut into two parts at the size mark of 25 kDa. Proteins > 25 kDa were transferred to a nitrocellulose membrane (0.2 µm, Trans-Blot Turbo, #1704158, Bio-Rad, Hercules, CA, USA). Proteins < 25 kDa were transferred to a PVDF membrane (0.2 µm, Trans-Blot Turbo, #1704156, Bio-Rad, Hercules, CA, USA). The membranes were blocked for 1h at RT in Odyssey blocking buffer (927-40000, LI-COR, Lincoln, NE, USA). Primary antibodies (rabbit anti-Bax, #5023, 1:500, Cell Signaling Technology, Danvers, MA, USA; internal control mouse β-Actin, #3700, 1:1000, Cell Signaling Technology, Danvers, MA, USA) diluted in Odyssey blocking buffer + 0.1% Tween (#P1379, Sigma-Aldrich, St. Louis, MO, USA) were incubated overnight at 4°C. The membranes were washed thrice for 5 min each with TBS-T (0.1% Tween) and then incubated with fluorescently labeled secondary antibodies (goat anti-mouse IRDye 680RD, #926-68070, LI-COR, Lincoln, NE, USA; goat anti-rabbit IRDye 800CW, #926-32211, LI-COR, Lincoln, NE, USA) in 0.1% Tween Odyssey blocking buffer for 1h at RT. For the PVDF membrane, 0.02% of SDS (Sigma-Aldrich, St. Louis, MO, USA) was added to the secondary antibody solution. Again, the membranes were washed thrice for 5 min each with TBS-T. The blots were scanned using the LI-COR Odyssey detection system (LI-COR, Lincoln, NE, USA).

### 4.11. Isolation of Regulatory B Cells (Bregs) from HNSCC Biopsy Samples

HNSCC biopsy samples were collected from patients of the University Hospital of Basel. The patients were scheduled for panendoscopy in general anesthesia as part of the clinical routine cancer staging. Carcinoma was confirmed by regular histopathological examination. Informed consent was obtained prior to the intervention and an additional biopsy was taken from the tumor tissue to be analyzed in our laboratory. The samples were dissociated with collagenase D and DNAse I. Immediately after digestion, cell surface staining for CD20, CD39, CD24, and CD38, their respective isotypes (see product information in Section 4.7), and compensation beads was performed. Cells were analyzed using a CytoFLEX flow cytometer (Beckman Coulter, Brea, CA, USA) and FlowJo Software version 10.6.2 for Mac. Breg markers were chosen based on previously published work by us or other colleagues [25,49,50].

### 4.12. Ethical Approval

A positive ethics opinion on the molecular biological examination of patient tissue in patients with head and neck tumors was obtained from the local ethics committee. Informed consent was obtained from all patients involved in the study.

Buffy coats of healthy individuals from the Swiss Red Cross Blood Donation Center Basel were collected after having obtained written informed consent from the donors.

### 4.13. Statistics

Statistical analysis was performed using GraphPad Prism version 9.0.0 for Mac, GraphPad Software, San Diego, CA, USA, www.graphpad.com, last accessed on 11 August 2022. All data are presented as means of at least three independent experiments ± SDs. Statistical methods for each analysis are described in detail in each figure legend or in the Results section. Depending on the experiment, the normalization/relative ratio was calculated as indicated. *p*-values ≤ 0.05 were considered significant.

## 5. Conclusions

It has been described that B cells, and particularly Breg subsets, make up an important part of the recruited immune cells or tumor-infiltrating immune cells and pivotally contribute to the pro-tumorigenic microenvironment [25,44,46].

Hereby, eATP and its main metabolite ADO are key constituents of the TME, crucially contributing as extracellular messengers to the tumor–host interaction. In the TME, eATP has displayed both pro- and anti-carcinogenic properties. The net effect of eATP and ADO in the TME or on immune cells is very complex and depends on a myriad of factors: eATP concentration; P2 receptor subtypes and their expression levels; presence and levels of ATP-degrading ectoenzymes, and other stimuli competing for the receptors or interfering with the signaling pathway [15,16,29]. For instance, other nucleotides such as urine diphosphate (UDP) and uridine triphosphate (UTP) compete for many P2Rs. In the TME, these nucleotides are highly enriched and recruit immune cells to the inflammatory site. UDP was reported to trigger anti-tumor activity by favoring TH1 and TH17 response over TH2 response and stimulating CD4+/CD8+ effector T-cell proliferation [73]. We also observed that TEX, as an additional external stimulus, interfered with the apoptotic signaling pathway of ATP (Figure 6, Figure 7, and Appendix A).

Generally, low ATP levels stimulate tumor proliferation and immunosuppression. High ATP levels activate tumor-infiltrating immune cells and support the anti-tumor response, also in conjunction with P2X7R-mediated toxicity. However, activation of the purinergic receptors P2YR and P2X7R is responsible for recruiting immune cells to the inflammatory site. Hence, highly immunosuppressive inflammatory cells are also recruited to the TME. Additionally, the function of the infiltrating immune cells can also be subverted by ATP and ADP in the TME to support tumor progression [15,16]. In HNSCC, it was reported that Tregs were actively recruited to the TME, where they contributed to the immune escape [74]. On the other hand, tumor-infiltrating effector cells, e.g., B, T, NK, and dendritic cells, were reported to be functionally impaired in HNSCC [75,76].

In addition to the nucleotides, TEX have been increasingly recognized as key players in cancer that are able to modulate immune cells. As previously described, TEX have shown dual effects in the TME [41,77,78,79]. TEX were reported to elicit immunostimulatory as well as immunosuppressive effects, translating into anti- and pro-tumorigenic effects [41,80,81,82]. In our study, we also observed two sides of the same coin: TEX had an immunostimulatory effect on B cells by downregulating the overall CD39 expression and reducing the CD39high B-cell subset; TEX had an immunoinhibitory effect by inhibiting B-cell proliferation. Additionally, the dual effect of TEX was also seen on various apoptosis-associated proteins. TEX upregulated pro- and anti-apoptotic proteins. This immunomodulatory influence of TEX seems to play a pivotal, yet also very complex, role in the TME.

New therapeutic agents targeting molecules expressed on B cells, such as CD39, CD73, and ADO receptors (especially A2A), are moving into early clinical trials. These treatments have a huge impact on B cells, and thus further work to improve the understanding of the underlying molecular interactions and implications of these therapies is required.

## Figures and Tables

**Figure 1 ijms-23-14446-f001:**
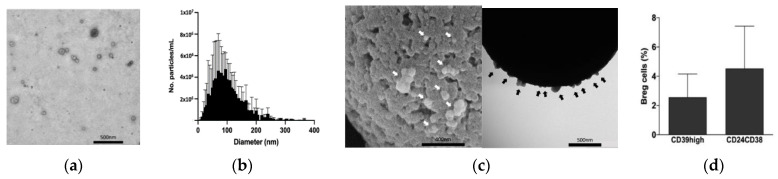
(**a**) Representative transmission electron microscope (TEM) image of exosomes isolated by size-exclusion chromatography and differential centrifugation from the HNSCC PCI-13 cell culture. (**b**) Size analysis of biological triplicates of four exosome isolates from the HNSCC PCI-13 cell culture shows the typical size range of exosomes (30–150 nm). Mean peak vesicle size with standard deviation: 89.47 ± 26.11 nm. Measurements were acquired using the ZetaView^®^ nanoparticle tracking analyzer (Particle Metrix). (**c**) Representative TEM images of Dynabeads^®^ with bound CD81+ exosomes (arrows) that were isolated from the HNSCC PCI-13 cell culture. Exosomes were isolated by size-exclusion chromatography and differential centrifugation, immunolabeled with anti-CD81 antibodies coupled to Dynabeads^®^, and imaged by TEM. (**d**) Mean of three flow cytometric experiments with standard deviation. A separate analysis of the CD39high and the double-positive CD24+/CD38+ B-cell populations after gating for CD20+ cells. Samples were from three different tumor samples of HNSCC patients.

**Figure 2 ijms-23-14446-f002:**
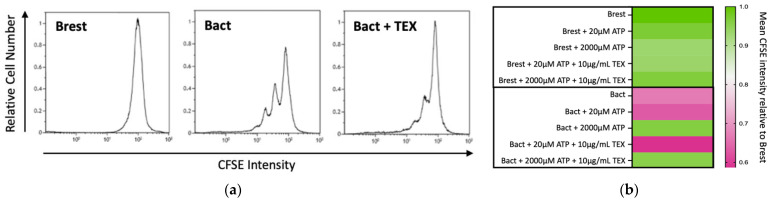
(**a**) Representative flow cytometry (CFSE assay) of resting B cells (Brest), activated B cells (Bact), and activated B cells co-incubated with 10 µg/mL of tumor-derived exosomes (TEX). (**b**) Heatmap with a relative proliferation index compared to Brest (value = 1) in a flow-cytometry-based CFSE assay. Lower values mean less CFSE fluorescence compared to Brest and, as such, proliferation of cells. The mean of four independent experiments is shown on the heatmap.

**Figure 3 ijms-23-14446-f003:**
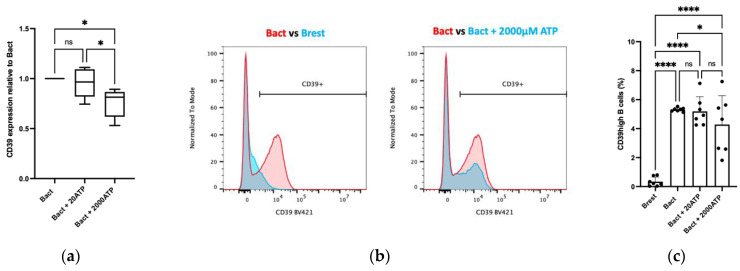
Flow cytometry data of CD39 expression on B cells in the presence of ATP (20 µM vs. 2000 µM). (**a**) Box plot graph illustrates the relative overall CD39 expression in activated B cells (Bact) and in the presence of ATP. The mean of six independent experiments with standard deviation is presented. The mean of each ATP intervention is related to Bact (value 1). The presence of 2000 µM ATP significantly decreased the CD39 expression in Bact. Bact vs. Bact + 2000 µM ATP: *p* * = 0.014; Bact + 20 µM ATP vs. Bact + 2000 µM ATP: *p* * = 0.012; ns = not significant: *p* > 0.05. (**b**) Activated B cells (Bact, red) showed a higher CD39 expression than resting B cells (Brest, blue). Bact with 2000 µM ATP (blue) showed a decreased CD39 expression. BV421 = Brilliant Violet 421. (**c**) Scatter dot bar charts of CD39high B cells (in %) after activation and in the presence of ATP. The mean of six independent experiments with standard deviation is presented. Activation of B cells increased the CD39high B-cell rate with high significance. Addition of 20 µM ATP did not impact the rate of CD39high B cells. The amount of 2000 µM ATP significantly decreased the rate of CD39high B cells. Brest vs. Bact: *p* **** < 0.0001; Brest vs. Bact + 20 µM ATP: *p* **** < 0.0001; Brest vs. Bact + 2000 µM ATP: *p* **** < 0.0001; Bact vs. Bact + 2000 µM ATP: *p* * = 0.047; ns = not significant: *p* > 0.05. Statistical significance was determined using a one-way ANOVA approach with multiple comparisons by controlling the false discovery rate.

**Figure 4 ijms-23-14446-f004:**
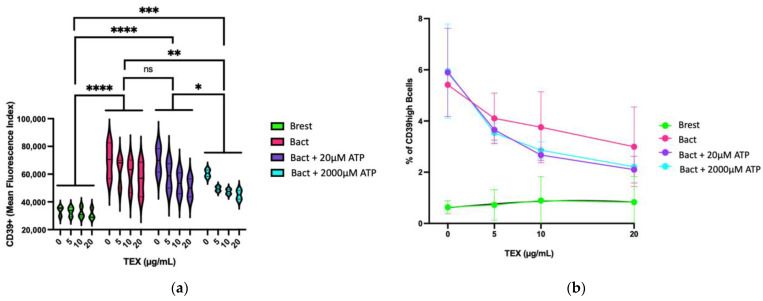
Flow cytometry data showing the dose-dependent influence of TEX on CD39 expression levels in activated B cells (Bact) or resting B cells (Brest) with 0, 20, or 2000 µM ATP added. (**a**) The graph shows the CD39 mean fluorescence index of the four groups. With high significance, activation of B cells increased the overall CD39 expression on B cells as compared to Brest (*p ***** < 0.0001). Further addition of 2000 µM ATP but not 20 µM ATP significantly reduced CD39 expression on Bact (*p *** = 0.0026). The influence of TEX within the individual groups indicated a trend toward reduced CD39 expression on B cells, especially in Bact. No change was seen in Brest. In Bact + 2000ATP co-incubated with the highest TEX concentration (20 µg/mL), CD39 expression approximated the range seen in Brest. Brest vs. Bact + 20 µM ATP: *p* **** < 0.0001; Brest vs. Bact + 2000 µM ATP: *p* *** = 0.0001; Bact + 20 µM ATP vs. Bact + 2000 µM ATP: *p* * = 0.0249; ns = not significant: *p* > 0.05. Statistical significance was calculated using a Holm–Sidak multiple comparisons test. (**b**) Dose titration curve of B cells with TEX measured by surface CD39 expression. The graph illustrates the dose-dependent reduction in CD39high B cells through TEX which is enhanced through ATP. At 20 µg/mL TEX, the CD39high rate in Bact with or without ATP dropped to almost the range measured in Brest. At this point, the difference between each Bact setting and Brest lost its significant value. With 0 µg/mL TEX: Brest vs. Bact: *p* *** = 0.0003; Brest vs. Bact + 20 µM ATP: *p* **** < 0.0001; Brest vs. Bact + 20 µM ATP *p* **** < 0.0001. With 5 µg/mL TEX: Brest vs. Bact: *p* *** = 0.0023; Brest vs. Bact + 20 µM ATP: *p* * = 0.0237; Brest vs. Bact + 20 µM ATP *p* * = 0.0323. With 10 µg/mL TEX: Brest vs. Bact: *p* *** = 0.0067. Statistical significance was calculated using a three-way ANOVA approach with Dunnett’s multiple comparisons test. The results of three independent experiments are presented.

**Figure 5 ijms-23-14446-f005:**
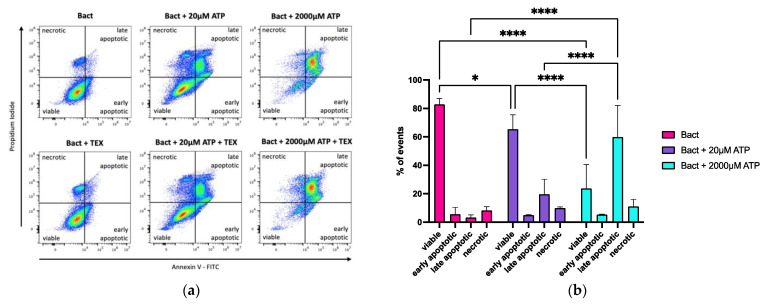
Flow cytometry analysis of annexin V and propidium iodide (PI) staining in activated B cells exposed to ATP and/or TEX. (**a**) Adding ATP to activated B cells (Bact) induced apoptosis in B cells, especially the high 2000 µM ATP dose (upper panel). TEX (10 µg/mL) did not appear to alter apoptosis or necrosis (lower panel). FITC = fluorescein isothiocyanate. (**b**) Bar charts showing the influence of ATP and TEX on cell death in B cells by annexin V and PI staining. The mean of three independent experiments with standard deviation is presented. ATP significantly reduced the viability of activated B cells and pushed them towards apoptosis (viable: Bact vs. Bact + 20 µM ATP: *p* * = 0.0384; Bact vs. Bact + 2000 µM ATP: *p* **** < 0.0001; Bact + 20 µM ATP vs. Bact + 2000 µM ATP: *p* **** < 0.0001; late apoptotic: Bact vs. Bact + 2000 µM ATP: *p* **** < 0.0001; Bact + 20 µM ATP vs. Bact + 2000 µM ATP: *p* **** < 0.0001). Adding TEX to either ATP concentration did not significantly alter the results (Appendix A). Statistical significance was determined using a two-way ANOVA with Tukey’s multiple comparisons test.

**Figure 6 ijms-23-14446-f006:**
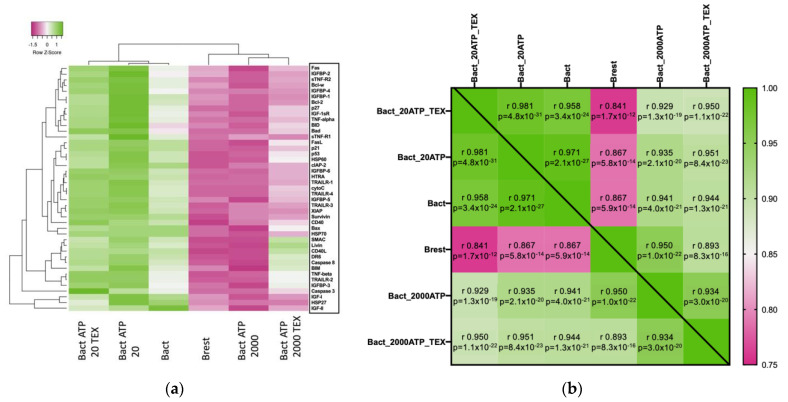
Heatmaps showing the protein expression of 43 different pro- and anti-apoptotic proteins of four independent experiments with isolated B cells in six different conditions (Brest: resting B cells; Bact: activated B cells; Bact with 20 µM or 2000 µM ATP; and Bact with 20 µM or 2000 µM ATP and TEX). (**a**) Unsupervised heatmap with the B cells in the six conditions on the x-axis and the 43 proteins on the *y*-axis. Unsupervised clustering uncovered two groups: the green-dominated group (Bact and Bact with 20 µM ATP with or without TEX) and the pink-dominated group (Brest and Bact with 2000 µM ATP with or without TEX). Distance measuring method: Pearson. Clustering method: average linkage. (**b**) Heatmap showing the protein expression as an overall mean. Differences between each group were highly significant. The most pronounced differences were between Brest and Bact or Bact + 20 µM ATP +/− TEX. Pearson correlation matrix with respective r-values (scale 0.75–1.00) and statistical significances (*p*-values).

**Figure 7 ijms-23-14446-f007:**
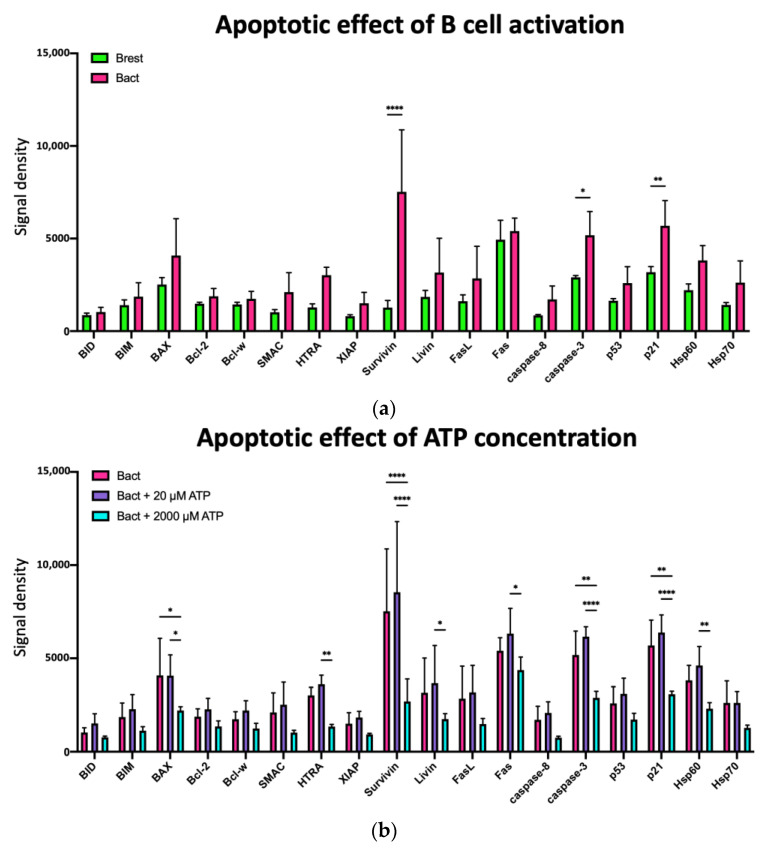
Bar charts comparing the expression of 18 selected apoptosis-associated proteins shown in Figure 6a. (**a**) Activated B cells (Bact) with a nonsignificant upregulation in 15 proteins compared to resting B cells (Brest). Significant upregulation for Survivin (*p* **** < 0.0001), Caspase 3 (*p* * = 0.0215), and p21 (*p* **= 0.0068). Significance determined by two-way ANOVA with Šídák’s multiple comparisons test. (**b**) Addition of 20 µM ATP to activated B cells (Bact) with nonsignificant impact on the expression of the selected 18 apoptosis-associated proteins. Addition of 2000 µM ATP to Bact with downregulation of all 18 of the pro- and anti-apoptotic proteins. Significant downregulation for B (Bact vs. Bact + 2000 µM ATP: *p* * = 0.0367; Bact + 20 µM ATP vs. Bact + 2000 µM ATP: *p* * = 0.0378), HTRA (Bact + 20 µM ATP vs. Bact + 2000 µM ATP: *p* ** = 0.0085), Survivin (Bact vs. Bact + 2000 ATP: *p* **** < 0.0001; Bact + 20 µM ATP vs. Bact + 2000 µM ATP: *p* **** < 0.0001), Livin (Bact + 20 µM ATP vs. Bact + 2000 µM ATP: *p* * = 0.0310), Fas (Bact + 20 µM ATP vs. Bact + 2000 µM ATP: *p* * = 0.0264), Caspase 3 (Bact vs. Bact + 2000 ATP: *p* ** = 0.0074; Bact + 20 µM ATP vs. Bact + 2000 µM ATP: *p* **** < 0.0001), p21 (Bact vs. Bact + 2000 ATP: *p* ** = 0.0018; Bact + 20 µM ATP vs. Bact + 2000 µM ATP: *p* **** < 0.0001), and Hsp60 (Bact + 20 µM ATP vs. Bact + 2000 µM ATP: *p* ** = 0.0068). Significance determined by two-way ANOVA with Tukey’s multiple comparisons test. (**c**) Addition of TEX to activated B cells (Bact) with 2000 µM ATP with a nonsignificant upregulation in 16 of the 18 apoptotic proteins. Significant upregulation for Survivin (Bact + 2000 µM ATP vs. Bact + 2000 µM ATP + TEX: *p* **** < 0.0001), and Livin (Bact + 2000 µM ATP vs. Bact + 2000 µM ATP + TEX: *p* * = 0.0362). Significance determined by two-way ANOVA with Tukey’s multiple comparisons test.

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
