# Peer review of "Differential Immunomodulatory Effects of Head and Neck Cancer-Derived Exosomes on B Cells in the Presence of ATP"

_ijms, 2022, doi:10.3390/ijms232214446_

Round 1
Reviewer 1 Report
This study by Coray et.al, studied the effect of cancer cell driven exosomes on B cells in the +/- ATP conditions and found their immunoinhibitory effects on B cells. This effect was further modulated in the presence of high concentration of ATP. The experiment performed support the conclusions drawn, however, there are multiple corrections and suggestions I have before this manuscript should be considered for a publication:
1. In Fig. 2a, does Y axis represent % Gated or number of cells?
2. In Fig. 2a, mention the amount of TEX taken for the CFSE assay
3. In Fig. 2b, group names are small and hard to read
4. In Fig. 3b, what is BV421, please mention this in the legend
5. In Fig. 3c legend: correct Scatter dot blot, should it not be a Column graph?
6. In Fig. 4b, Statistical values must be present on graph or in the legend, not in the supplemental. In the figure legend, mention µM before each ATP concentration rather saying 20ATP or 2000ATP
7. In Fig 5a, I would suggest showing this data by dot plot including Annexin and PI double positive cells, which in the literature are known as late apoptotic cells and would be important to see if early apoptotic Annexin+ cells are finally going into death pathway?
8. In Fig 6a, did authors include Brest+ TEX group to see the effect of TEX on these protein in resting B cells to compare with other data??
9. Line 506: remove underline from “specific” word
10. In Fig 7a, what does intensity means at “Y” axis? The figure panel names should on the top left at each panel.
Author Response
Please see the attachment. Thank you so much already in advance.

Reviewer 2 Report
In this study, the authors aim to evaluate the effect of HNSCC-derived exosomes in the presence of high levels of ATP on B cells in order to investigate the interaction between TME and the immune system. Some major revisions need to be done before acceptance:
1) General Revision:
-Typography: the authors should read thoroughly their manuscript and check: 1) space between words; 2) English of some sentences
2) Introduction:
-I suggest to better explain in the introduction section the double immunostimulatory and immunosuppressive effect of TEX in TME found in the literature and citated in the discussion in order to make the obtained results clearer.
3) Material and Methods:
-Please, add more references to justify the methods of isolation of peripheral blood B-cells and isolation of regulatory B cells (Bregs) from HNSCC biopsy samples.
-Please, specify in the Western Blotting section the name, the code, commercial brand, city and country of provenance for all antibodies and materials used.
4) Results:
-I suggest to add the original blots of all proteins analyzed by western blot.
Author Response

(The authors gave the same response as above.)

Round 2
Reviewer 1 Report
Authors have addressed all my comments and manuscript can be considered for a publication.
Author Response
Dear Reviewer,
Firstly, we would like to thank you for your valuable time and your kind review. We are very happy that we could fulfill your expectations. Our manuscript was submitted to the English Editing Service of MDPI to improve our English language. Again, thank you very much for your kind review.
Reviewer 2 Report
The manuscript is well written although some things need to be improved. Particularly, I suggest to increase the Figures resolution and to performe a revision of English language
Author Response
Dear Reviewer,
Firstly, we would like to thank you for your valuable time and your kind review. We are very happy that we could broadly fulfill your expectations. Our manuscript was submitted to the English Editing Service of MDPI to improve our English language. We are very sorry about the resolution. It seems that the conversion of word to pdf created the loss of resolution. We are trying to fix this problem. Again, thank you very much for your kind review.